# HUMAN PERCEPTION IN COMPUTER VISION / CONFERENCE SUBMISSIONS

**Ron Dekel** *
Department of Neurobiology
Weizmann Institute of Science
Rehovot, PA 7610001, Israel
`ron.dekel@weizmann.ac.il`

## ABSTRACT

Computer vision has made remarkable progress in recent years. Deep neural network (DNN) models optimized to identify objects in images exhibit unprecedented task-trained accuracy and, remarkably, some generalization ability: new visual problems can now be solved more easily based on previous learning. Biological vision (learned in life and through evolution) is also accurate and general-purpose. Is it possible that these different learning regimes converge to similar problem-dependent optimal computations? We therefore asked whether the human system-level computation of visual perception has DNN correlates and considered several anecdotal test cases. We found that perceptual sensitivity to image changes has DNN mid-computation correlates, while sensitivity to segmentation, crowding and shape has DNN end-computation correlates. Our results quantify the applicability of using DNN computation to estimate perceptual loss, and are consistent with the fascinating theoretical view that properties of human perception are a consequence of architecture-independent visual learning.

## 1 QUICK EXPERT SUMMARY

Considering the learned computation of ImageNet-trained DNNs, we find:

- Large computation changes for perceptually salient image changes (Figure 1).
- Gestalt: segmentation, crowding, and shape interactions in computation (Figure 2).
- Contrast constancy: bandpass transduction in first layers is later corrected (Figure 3).

These properties are reminiscent of human perception, perhaps because learned general-purpose classifiers (human and DNN) tend to converge.

## 2 INTRODUCTION

Deep neural networks (DNNs) are a class of computer learning algorithms that have become widely used in recent years (LeCun et al., 2015). By training with millions of examples, such models achieve unparalleled degrees of task-trained accuracy (Krizhevsky et al., 2012). This is not unprecedented on its own - steady progress has been made in computer vision for decades, and to some degree current designs are just scaled versions of long-known principles (Lecun et al., 1998). In previous models, however, only the design is general-purpose, while learning is mostly specific to the context of a trained task. Interestingly, for current DNNs trained to solve a large-scale image recognition problem (Russakovsky et al., 2014), the learned computation is useful as a building block for drastically different and untrained visual problems (Huh et al., 2016; Yosinski et al., 2014).

For example, orientation- and frequency-selective features (Gabor patches) can be considered general-purpose visual computations. Such features are routinely discovered by DNNs (Krizhevsky et al., 2012; Zeiler & Fergus, 2013), by other learning algorithms (Hinton & Salakhutdinov, 2006;

---

*`https://sites.google.com/site/rondekelhomepage/`

Lee et al., 2008; 2009; Olshausen & Field, 1997), and are extensively hard-coded in computer vision (Jain & Farrokhnia, 1991). Furthermore, a similar computation is believed to underlie the spatial response properties of visual neurons of diverse animal phyla (Carandini et al., 2005; DeAngelis et al., 1995; Hubel & Wiesel, 1968; Seelig & Jayaraman, 2013), and is evident in human visual perception (Campbell & Robson, 1968; Fogel & Sagi, 1989; Neri et al., 1999). This diversity culminates in satisfying theoretical arguments as to why Gabor-like features are so useful in general-purpose vision (Olshausen, 1996; Olshausen & Field, 1997).

As an extension, general-purpose computations are perhaps of universal use. For example, a dimensionality reduction transformation that optimally preserves recognition-relevant information may constitute an ideal computation for both DNN and animal. More formally, different learning algorithms with different physical implementations may converge to the same computation when similar (or sufficiently general) problems are solved near-optimally. Following this line of reasoning, DNN models with good general-purpose computations may be computationally similar to biological visual systems, even more so than less accurate and less general biologically plausible simulations (Kriegeskorte, 2015; Yamins & DiCarlo, 2016).

Related work seems to be consistent with computation convergence. First, different DNN training regimes seem to converge to a similar learned computation (Li et al., 2015; Zhou et al., 2014). Second, image representation may be similar in trained DNN and in biological visual systems. That is, when the same images are processed by DNN and by humans or monkeys, the final DNN computation stages are strong predictors of human fMRI and monkey electrophysiology data collected from visual areas V4 and IT (Cadieu et al., 2014; Khaligh-Razavi & Kriegeskorte, 2014; Yamins et al., 2014). Furthermore, more accurate DNN models exhibit stronger predictive power (Cadieu et al., 2014; Dubey & Agarwal, 2016; Yamins et al., 2014), and the final DNN computation stage is even a strong predictor of human-perceived shape discrimination (Kubilius et al., 2016). However, some caution is perhaps unavoidable, since measured similarity may be confounded with categorization consistency, view-invariance resilience, or similarity in the inherent difficulty of the tasks undergoing comparison. A complementary approach is to consider images that were produced by optimizing trained DNN-based perceptual metrics (Gatys et al., 2015a;b; Johnson et al., 2016; Ledig et al., 2016), which perhaps yields undeniable evidence of non-trivial computational similarity, although a more objective approach may be warranted.

Here, we quantify the similarity between human visual perception, as measured by psychophysical experiments, and individual computational stages (layers) in feed-forward DNNs trained on a large-scale image recognition problem (ImageNet LSVRC). Comparison is achieved by feeding the experimental image stimuli to the trained DNN and comparing a DNN metric (mean mutual information or mean absolute change) to perceptual data. The use of reduced (simplified and typically non-natural) stimuli ensures identical inherent task difficulty across compared categories and prevents confounding of categorization consistency with measured similarity. Perception, a system-level computation, may be influenced less by the architectural discrepancy (biology vs. DNN) than are neural recordings.

## 3 CORRELATE FOR IMAGE CHANGE SENSITIVITY

From a perceptual perspective, an image change of fixed size has different saliency depending on image context (Polat & Sagi, 1993). To investigate whether the computation in trained DNNs exhibits similar contextual modulation, we used the Local Image Masking Database (Alam et al., 2014), in which 1080 partially-overlapping images were subjected to different levels of the same random additive noise perturbation, and for each image, a psychophysical experiment determined the threshold noise level at which the added-noise image is discriminated from two noiseless copies at $75\%$ (Figure 1a). Threshold is the objective function that is compared with an $L_1$-distance correlate in the DNN representation. The scale of measured threshold was:

$$20 \cdot \log_{10} \left( \frac{\text{std}\,(noise)}{T} \right),$$ (1)

where $\text{std}\,(noise)$ is the standard deviation of the additive noise, and $T$ is the mean image pixel value calculated over the region where the noise is added (i.e. image center).

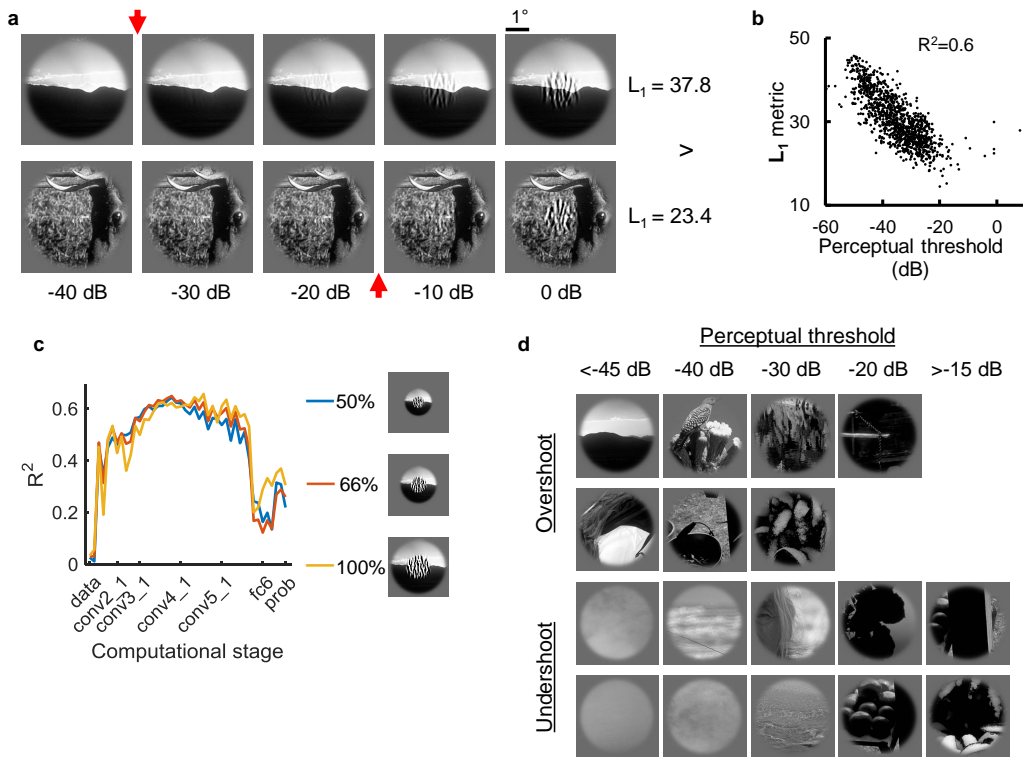

Figure 1: Predicting perturbation thresholds. **a**, For a fixed image perturbation, perceptual detection threshold (visualized by red arrow) depends on image context. **b**, Measured perceptual threshold is correlated with the average $L_1$ change in DNN computation due to image perturbation (for DNN model VGG-19, image scale=100%). **c**, Explained variability ($R^2$) of perceptual threshold data when $L_1$ change is based on isolated computational layers for different input image scales. Same VGG-19 model as in (b). X-axis labels: `data` refers to raw image pixel data, `conv*_1` and `fc_*` are the before-ReLU output of a convolution and a fully-connected operation, respectively, and `prob` is the output class label probabilities vector. **d**, Example images for whcih predicted threshold in b is much higher than perceptually measured ("Overshoot", where perturbation saliency is better than predicted), or vise versa ("Undershoot"). Examples are considered from several perceptual threshold ranges ($\pm 2$ dB of shown number).

The DNN correlate of perceptual threshold we used was the average $L_1$ change in DNN computation between added-noise images and the original, noiseless image. Formally,

$$L_1^{i,n}(I) = \left| \overline{a_i\left(I + noise\left(n\right)\right) - a_i\left(I\right)} \right|, \qquad (2)$$

where $a_i(X)$ is the activation value of neuron $i$ during the DNN feedforward pass for input image $X$, and the inner average (denoted by bar) is taken over repetitions with random $n$-sized noise (noise is introduced at random phase spectra in a fixed image location, an augmentation that follows the between-image randomization described by Alam et al., 2014; the number of repetitions was 10 or more). Unless otherwise specified, the final $L_1$ prediction is $L_1^{i,n}$ averaged across noise levels ($-40$ to 25 dB with 5-dB intervals) and computational neurons (first within and then across computational stages). Using $L_1$ averaged across noise levels as a correlate for the noise level of perceptual threshold is a simple approximation with minimal assumptions.

Results show that the $L_1$ metric is correlated with the perceptual threshold for all tested DNN architectures (Figure 1b, 4a-c). In other words, higher values of the $L_1$ metric (indicating larger changes in DNN computation due to image perturbation, consistent with higher perturbation saliency) are

| Model | $R^2$ | SROCC | RMSE |
|---|---|---|---|
| Signal-noise ratio | .20 | .39 | 7.67 |
| Spectral change | .25 | .61 | 7.42 |
| RMS contrast (Alam et al., 2014) | .27* | .46 | - |
| $L_1$ VGG-19 (50%) | .57 | .77 | 5.57 |
| $L_1$ VGG-19 (66%) | .60 | .79 | 5.42 |
| **$L_1$ VGG-19 (100%)** | **.60** | **.79** | **5.40** |
| Perceptual model** (Alam et al., 2014) | **.60*** | .70 | 5.73 |
| Inter-person (Alam et al., 2014) | .84* | .87 | 4.08 |

Table 1: Prediction accuracy. Percent of linearly explained variability ($R^2$), absolute value of Spearman rank-order correlation coefficient (SROCC), and the root mean squared error of the linear prediction (RMSE) are presented for each prediction model. Note the measurement scale of the threshold data being predicted (Eq. 1). (*) Thresholds linearized through a logistic transform before prediction (see Larson & Chandler, 2010), possibly increasing but not decreasing measured predictive strength. (**) Average of four similar alternatives.

correlated with lower values of measured perceptual threshold (indicating that weaker noise levels are detectable, i.e. higher saliency once more).

To quantify and compare predictive power, we considered the percent of linearly explained variability ($R^2$). For all tested DNN architectures, the prediction explains about $60\%$ of the perceptual variability (Tables 1, 2; baselines at Tables 3-5), where inter-person similarity representing theoretical maximum is 84% (Alam et al., 2014). The DNN prediction is far more accurate than a prediction based on simple image statistical properties (e.g. RMS contrast), and is on par with a detailed perceptual model that relies on dozens of psychophysically collected parameters (Alam et al., 2014). The Spearmann correlation coefficient is much higher compared with the perceptual model (with an absolute SROCC value of about 0.79 compared with 0.70, Table 1), suggesting that the $L_1$ metric gets the order right but not the scale. We did not compare these results with models that fit the experimental data (e.g. Alam et al., 2015; Liu & Allebach, 2016), since the $L_1$ metric has no explicit parameters. Also, different DNN architectures exhibited high similarity in their predictions ($R^2$ of about 0.9, e.g. Figure 4d).

Prediction can also be made from isolated computational stages, instead of across all stages as before. This analysis shows that the predictive power peaks mid-computation across all tested image scales (Figure 1c). This peak is consistent with use of middle DNN layers to optimize perceptual metrics (Gatys et al., 2015a;b; Ledig et al., 2016), and is reminiscent of cases in which low- to mid-level vision is the performance limiting computation in the detection of at-threshold stimuli (Campbell & Robson, 1968; Del Cul et al., 2007).

Finally, considering the images for which the $L_1$-based prediction has a high error suggests a factor which causes a systematic inconsistency with perception (Figures 1d, 6). This factor may be related to the mean image luminance: by introducing noise perturbations according to the scale of Equation 1, a fixed noise size (in dB) corresponds to smaller pixel changes in dark compared with bright images. (Using this scales reflects an assumption of multiplicative rather than additive conservation; this assumption may be justified for the representation at the final but perhaps not the intermediate computational stages considering the log-linear contrast response discussed in Section 5). Another factor may the degree to which image content is identifiable.

# 4 CORRELATE FOR MODULATION OF SENSITIVITY BY CONTEXT

The previous analysis suggested gross computational similarity between human perception and trained DNNs. Next, we aimed to extend the comparison to more interpretable properties of perception by considering more highly controlled designs. To this end, we considered cases in which a static background context modulates the difficulty of discriminating a foreground shape, despite no spatial overlap of foreground and background. This permits interpretation by considering the cause of the modulation.

We first consider segmentation, in which arrangement is better discriminated for arrays of consistently oriented lines compared with inconsistently oriented lines (Figure 2a) (Pinchuk-Yacobi et al., 2016). Crowding is considered next, where surround clutter that is similar to the discriminated target leads to deteriorated discrimination performance (Figure 2b) (Livne & Sagi, 2007). Last to be addressed is object superiority, in which a target line location is better discriminated when it is in a shape-forming layout (Figure 2c) (Weisstein & Harris, 1974). In this case, clutter is controlled by having the same fixed number of lines in context. To measure perceptual discrimination, these works introduced performance-limiting manipulations such as location jittering, brief presentation, and temporal masking. While different manipulations showed different measured values, order-of-difficulty was typically preserved. Here we changed all the original performance-limiting manipulations to location jittering (whole-shape or element-wise, see Section 8.4).

To quantify discrimination difficulty in DNNs, we measured the target-discriminative information of isolated neurons (where performance is limited by location jittering noise), then averaged across all neurons (first within and then across computational layer stages). Specifically, for each neuron, we measured the reduction in categorization uncertainty due to observation, termed mutual information (MI):

$$MI\left(A_i; C\right) = H\left(C\right) - H\left(C|A_i\right),\qquad(3)$$

where H stands for entropy, and $A_i$ is a random variable for the value of neuron i when the DNN processes a random image from a category defined by the random variable C. For example, if a neuron gives a value in the range of $100.0$ to $200.0$ when the DNN processes images from category A, and $300.0$ to $400.0$ for category B, then the category is always known by observing the value, and so mutual information is high (MI=1 bits). On the other extreme, if the neuron has no discriminative task information, then MI=0 bits. To measure MI, we quantized activations into eight equal-amount bins, and used 500 samples (repetitions having different location jittering noise) across categories. The motivation for this correlate is the assumption that the perceptual order-of-difficulty reflects the quantity of task-discriminative information in the representation.

Results show that, across hundreds of configurations (varying pattern element size, target location, jitter magnitude, and DNN architecture; see Section 8.4), the qualitative order of difficulty in terms of the DNN MI metric is consistent with the order of difficulty measured in human psychophysical experiments, for the conditions addressing segmentation and crowding (Figures 2d, 7; for baseline models see Figure 8). It is interesting to note that the increase in similarity develops gradually along different layer types in the DNN computation (i.e. not just pooling layers), and is accompanied by a gradual increase in the quantity of task-relevant information (Figure 2e-g). This indicates a link between task relevance and computational similarity for the tested conditions. Note that unlike the evident increase in isolated unit task information, the task information from all units combined decreases by definition along any computational hierarchy. An intuition for this result is that the total hidden information decreases, while more accessible per-unit information increases.

For shape formation, four out of six shapes consistently show order of difficulty like perception, and two shapes consistently do no (caricature at Figure 2h; actual data at Figure 9).

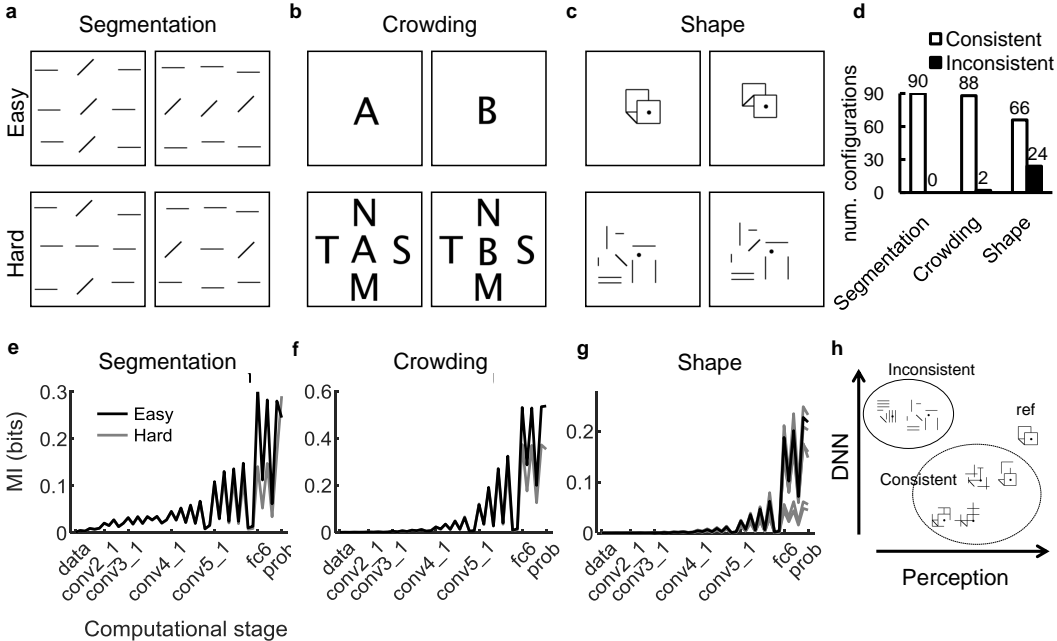

Figure 2: Background context. **a-c**, Illustrations of reproduced discrimination stimuli for three psychophysical experiments (actual images used were white-on-black rather than black-on-white, and pattern size was smaller, see Figures 12-14). **d**, Number of configurations for which order-of-difficulty in discrimination is qualitatively consistency with perception according to a mutual information DNN metric. Configurations vary in pattern (element size, target location, and jitter magnitude; see Section 8.4) and in DNN architecture used (CaffeNet, GoogLeNet, VGG-19, and ResNet-152). DNN metric is the average across neurons of the isolated neuron target-discriminative information (averaged first within, and then across computational layer stages), where performance is limited by location jittering (e.g. evident jitter in illustrations). **e-g**, The value of the MI metric across computational layers of model VGG-19 for a typical pattern configuration. The six "hard" (gray) lines in Shape MI correspond to six different layouts (see Section 8.4.3). Analysis shows that for isolated computation stages, similarity to perception is evident only at the final DNN computation stages. **h**, A caricature summarizing the similarity and discrepancy of perception and the MI-based DNN prediction for Shape (see Figure 9).

## 5 CORRELATE FOR CONTRAST SENSITIVITY

A cornerstone of biological vision research is the use of sine gratings at different frequencies, orientations, and contrasts (Campbell & Robson, 1968). Notable are results showing that the lowest perceivable contrast in human perception depends on frequency. Specifically, high spatial frequencies are attenuated by the optics of the eye, and low spatial frequencies are believed to be attenuated due to processing inefficiencies (Watson & Ahumada, 2008), so that the lowest perceivable contrast is found at intermediate frequencies. (To appreciate this yourself, examine Figure 3a). Thus, for low-contrast gratings, the physical quantity of contrast is not perceived correctly: it is not preserved across spatial frequencies. Interestingly, this is corrected for gratings of higher contrasts, for which perceived contrast is more constant across spatial frequencies (Georgeson & Sullivan, 1975).

The DNN correlate we considered is the mean absolute change in DNN representation between a gray image and sinusoidal gratings, at all combinations of spatial frequency and contrast. Formally, for neurons in a given layer, we measured:

$$L_1(contrast, frequency) = \frac{1}{N_{neurons}} \sum_{i=1}^{N_{neurons}} \left| \overline{a_i\left(contrast, frequency\right)} - a_i\left(0,0\right) \right|, \quad (4)$$

where $\overline{a_i\left(contrast, frequency\right)}$ is the average activation value of neuron $i$ to 250 sine images (random orientation, random phase), $a_i\left(0,0\right)$ is the response to a blank (gray) image, and $N_{neurons}$ is the number of neurons in the layer. This measure reflects the overall change in response vs. the gray image.

Results show a bandpass response for low-contrast gratings (blue lines strongly modulated by frequency, Figures 3, 10), and what appears to be a mostly constant response at high contrast for end-computation layers (red lines appear more invariant to frequency), in accordance with perception.

We next aimed to compare these results with perception. Data from human experiments is generally iso-output (i.e. for a pre-set output, such as 75% detection accuracy, the input is varied to find the value which produce the preset output). However, the DNN measurements here are iso-input (i.e. for a fixed input contrast the $L_1$ is measured). As such, human data should be compared to the interpoalted inverse of DNN measurements. Specifically, for a set output value, the interpolated contrast value which produce the output is found for every frequency (Figure 11). This analysis permits quantifying the similarity of iso-output curves for human and DNN, measured here as the percent of log-Contrast variability in human measurements which is explained by the DNN predictions. This showed a high explained variability at the end computation stage (prob layer, $R^2 = 94\%$), but importantly, a similarly high value at the first computational stage (conv1_1 layer, $R^2 = 96\%$). Intiutively, while the "internal representation" variability in terms of $L_1$ is small, the iso-output number-of-input-contrast-cahnges variability is still high. For example. for the prob layer, about the same $L_1$ is measured for (Contrast=1,freq=75) and for (Contrast=0.18,freq=12).

An interesting, unexpected observation is that the logarithmically spaced contrast inputs are linearly spaced at the end-computation layers. That is, the average change in DNN representation scales logarithmically with the size of input change. This can be quantified by the correlation of output $L_1$ with log Contrast input, which showed $R^2 = 98\%$ (averaged across spatial frequencies) for prob, while much lower values were observed for early and middle layers (up to layer fc7). The same computation when scrambling the learned parameters of the model showed $R^2 = 60\%$. Because the degree of log-linearity observed was extremely high, it may be an important emergent property of the learned DNN computation, which may deserve further investigation. However, this property is only reminiscent and not immediately consistent with the perceptual power-law scaling (Gottesman et al., 1981).

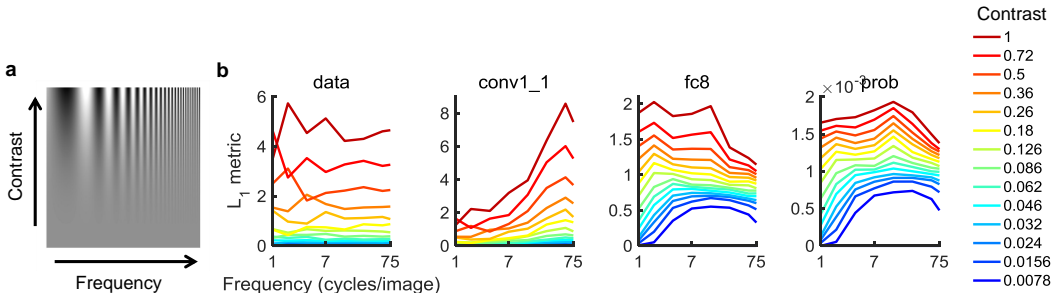

Figure 3: Contrast sensitivity. **a**. Perceived contrast is strongly affected by spatial frequency at low contrast, but less so at high contrast (which preserves the physical quantity of contrast and thus termed constancy). **b**. The $L_1$ change in VGG-19 representation between a gray image and images depicting sinusoidal gratings at each combination of sine spatial frequency (x-axis) and contrast (color) (random orientation, random phase), considering the raw image pixel data representation (data), the before-ReLU output of the first convolutional layer representation (conv1_1), the output of the last fully-connected layer representation (fc8), and the output class label probabilities representation (prob).

# 6 DISCUSSION

## 6.1 HUMAN PERCEPTION IN COMPUTER VISION

It may be tempting to believe that what we see is the result of a simple transformation of visual input. Centuries of psychophysics have, however, revealed complex properties in perception, by crafting stimuli that isolate different perceptual properties. In our study, we used the same stimuli to investigate the learned properties of deep neural networks (DNNs), which are the leading computer vision algorithms to date (LeCun et al., 2015).

The DNNs we used were trained in a supervised fashion to assign labels to input images. To some degree, this task resembles the simple verbal explanations given to children by their parents. Since human perception is obviously much richer than the simple external supervision provided, we were not surprised to find that the best correlate for perceptual saliency of image changes is a part of the DNN computation that is only supervised indirectly (i.e. the mid-computation stage). This similarity is so strong, that even with no fine-tuning to human perception, the DNN metric is competitively accurate, even compared with a direct model of perception.

This strong, quantifiable similarity to a gross aspect of perception may, however, reflect a mix of similarities and discrepancies in different perceptual properties. To address isolated perceptual effects, we considered experiments that manipulate a spatial interaction, where the difficulty of discriminating a foreground target is modulated by a background context. Results showed modulation of DNN target diagnostic, isolated unit information, consistent with the modulation found in perceptual discrimination. This was shown for contextual interactions reflecting grouping/segmentation (Harris et al., 2015), crowding/clutter (Livne & Sagi, 2007; Pelli et al., 2004), and shape superiority (Weisstein & Harris, 1974). DNN similarity to these groupings/gestalt phenomena appeared at the end-computation stages.

No less interesting, are the cases in which there is no similarity. For example, perceptual effects related to 3D (Erdogan & Jacobs, 2016) and symmetry (Pramod & Arun, 2016) do not appear to have a strong correlate in the DNN computation. Indeed, it may be interesting to investigate the influence of visual experience in these cases. And, equally important, similarity should be considered in terms of specific perceptual properties rather than as a general statement.

## 6.2 RECURRENT VS. FEEDFORWARD CONNECTIVITY

In the human hierarchy of visual processing areas, information is believed to be processed in a feed-forward sweep, followed by recurrent processing loops (top-down and lateral) (Lamme & Roelfsema, 2000). Thus, for example, the early visual areas can perform deep computations. Since mapping from visual areas to DNN computational layers is not simple, it will not be considered here. (Note that ResNet connectivity is perhaps reminiscent of unrolled recurrent processing).

Interestingly, debate is ongoing about the degree to which visual perception is dependent on recurrent connectivity (Fabre-Thorpe et al., 1998; Hung et al., 2005): recurrent representations are obviously richer, but feedforward computations converge much faster. An implicit question here regarding the extent of feasible feed-forward representations is, perhaps: Can contour segmentation, contextual influences, and complex shapes be learned? Based on the results reported here for feed-forward DNNs, a feedforward representation may seem sufficient. However, the extent to which this is true may be very limited. In this study we used small images with a small number of lines, while effects such as contour integration seem to take place even in very large configurations (Field et al., 1993). Such scaling seems more likely in a recurrent implementation. As such, a reasonable hypothesis may be that the full extent of contextual influence is only realizable with recurrence, while feedforward DNNs learn a limited version by converging towards a useful computation.

## 6.3 IMPLICATIONS AND FUTURE WORK

### 6.3.1 USE IN BRAIN MODELING

The use of DNNs in modeling of visual perception (or of biological visual systems in general) is subject to a tradeoff between accuracy and biological plausibility. In terms of architecture, other deep models better approximate our current understanding of the visual system (Riesenhuber &

Poggio, 1999; Serre, 2014). However, the computation in trained DNN models is quite general-purpose (Huh et al., 2016; Yosinski et al., 2014) and offers unparalleled accuracy in recognition tasks (LeCun et al., 2015). Since visual computations are, to some degree, task- rather than architecture-dependent, an accurate and general-purpose DNN model may better resemble biological processing than less accurate biologically plausible ones (Kriegeskorte, 2015; Yamins & DiCarlo, 2016). We support this view by considering a controlled condition in which similarity is not confounded with task difficulty or categorization consistency.

### 6.3.2 USE IN PSYCHOPHYSICS

Our results imply that trained DNN models have good predictive value for outcomes of psychophysical experiments, permitting a zero-cost first-order approximation. Note, however, that the scope of such simulations may be limited, since learning (Sagi, 2011) and adaptation (Webster, 2011) were not considered here.

Another fascinating option is the formation of hypotheses in terms of mathematically differentiable trained-DNN constraints, whereby it is possible to efficiently solve for the visual stimuli that optimally dissociate the hypotheses (see Gatys et al. 2015a;b; Mordvintsev et al. 2015 and note Goodfellow et al. 2014; Szegedy et al. 2013). The conclusions drawn from such stimuli can be independent of the theoretical assumptions about the generating process (for example, creating new visual illusions that can be seen regardless of how they were created).

### 6.3.3 USE IN ENGINEERING (A PERCEPTUAL LOSS METRIC)

As proposed previously (Dosovitskiy & Brox, 2016; Johnson et al., 2016; Ledig et al., 2016), the saliency of small image changes can be estimated as the representational distance in trained DNNs. Here, we quantified this approach by relying on data from a controlled psychophysical experiment (Alam et al., 2014). We found the metric to be far superior to simple image statistical properties, and on par with a detailed perceptual model (Alam et al., 2014). This metric can be useful in image compression, whereby optimizing degradation across image sub-patches by comparing perceptual loss may minimize visual artifacts and content loss.

### ACKNOWLEDGMENTS

We thank Yoram Bonneh for his valuable questions which led to much of this work.

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

# 7 APPENDIX: FIGURES AND TABLES

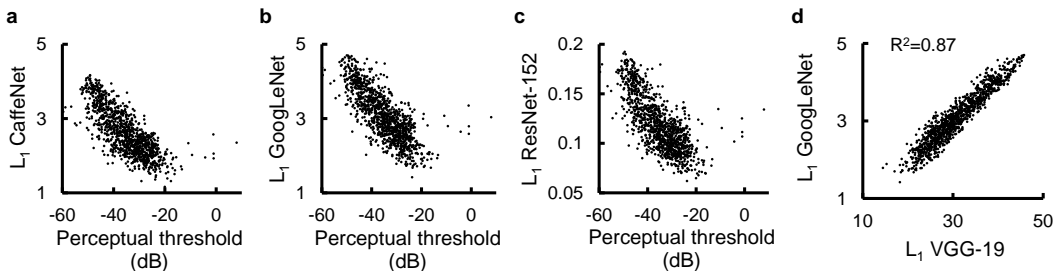

Figure 4: Predicting perceptual sensitivity to image changes (following Figure 1). **a-c**, The $L_1$ change in CaffeNet, GoogLeNet, and ResNet-152 DNN architectures as a function of perceptual threshold. **d**, The $L_1$ change in GoogLeNet as a function of the $L_1$ change in VGG-19.

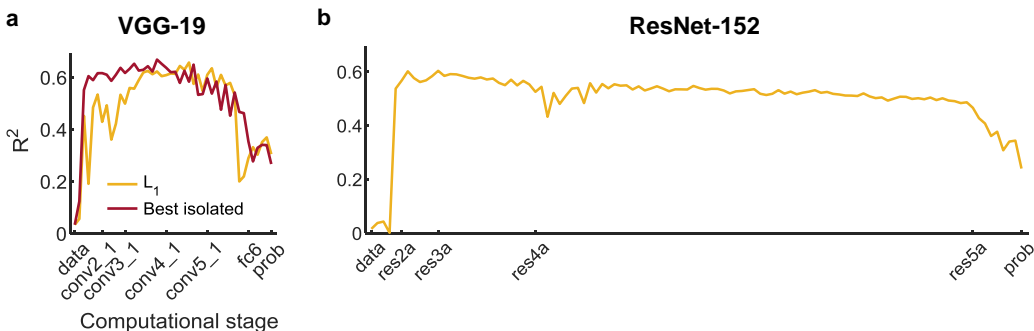

Figure 5: Prediction accuracy as a function of computational stage. **a**, Predicting perceptual sensitivity for model VGG-19 using the best single kernel (i.e. using one fitting parameter, no cross validation), vs. the standard $L_1$ metric (reproduced from Figure 1). **b**, For non-branch computational stages of model ResNet-152.

| Model | $R^2$ | SROCC | RMSE | Recognition accuracy |
|---|---|---|---|---|
| CaffeNet | .59 | .78 | 5.44 | 56% |
| GoogLeNet | .59 | .79 | 5.45 | 66% |
| VGG-19 | .60 | .79 | 5.40 | 70% |
| ResNet-152 | .53 | .74 | 5.82 | 75% |

Table 2: Accuracy of perceptual sensitivity prediction and task-trained ImageNet center-crop top-1 validation accuracy for different DNN models (following Table 1 from which third row is reproduced; used scale: 100%). The quality of prediction for ResNet-152 improves dramatically if only the first tens of layers are considered (see Figure 5b).

| Model | $R^2$ | SROCC | RMSE |
|---|---|---|---|
| VGG-19, scrambled weights | .18 | .39 | 7.76 |
| Gabor filter bank | .32 | .12 | 8.03 |
| Steerable-pyramid filter bank | .37 | .15 | 7.91 |

Table 3: Accuracy of perceptual sensitivity prediction for baseline models (see Section 8.2; used scale: 100%).

| Model | $R^2$ | SROCC | RMSE | Recognition accuracy |
|---|---|---|---|---|
| CaffeNet iter 1 | .46 | .67 | 6.30 | 0% |
| CaffeNet iter 50K | .59 | .79 | 5.43 | 37% |
| CaffeNet iter 100K | .60 | .79 | 5.41 | 39% |
| CaffeNet iter 150K | .60 | .78 | 5.43 | 53% |
| CaffeNet iter 200K | .59 | .78 | 5.45 | 54% |
| CaffeNet iter 250K | .59 | .78 | 5.43 | 56% |
| CaffeNet iter 300K | .59 | .78 | 5.44 | 56% |
| CaffeNet iter 310K | .59 | .78 | 5.44 | 56% |

Table 4: Accuracy of perceptual sensitivity prediction during CaffeNet model standard training (used scale: 100%). Last row reproduced from Table 2.

| Scale | Metric | Augmentation | Noise range | $R^2$ | SROCC | RMSE |
|---|---|---|---|---|---|---|
| **100%** | $L_1$ | noise phase | -40:25 dB | .60 | .79 | 5.40 |
| **66%** | $L_1$ | noise phase | -40:25 dB | .60 | .79 | 5.42 |
| **50%** | $L_1$ | noise phase | -40:25 dB | .57 | .77 | 5.57 |
| 100% | **$L_2$** | noise phase | -40:25 dB | .62 | .80 | 5.29 |
| 100% | $L_1$ | **None** | -40:25 dB | .58 | .77 | 5.55 |
| 100% | $L_1$ | noise phase | **-20**:25 dB | .59 | .78 | 5.46 |
| 100% | $L_1$ | noise phase | -40:**5** dB | .59 | .79 | 5.43 |

Table 5: Robustness of perceptual sensitivity prediction for varying prediction parameters for model VGG-19. First three rows reproduced from Table 1. Measurements for the lower noise range of -60:-40 dB were omitted by mistake.

| Model | Day 1 | Days 2-4 | Masked |
|---|---|---|---|
| VGG-19 | **.36** | **.37** | .15 |
| GoogLeNet | .31 | .22 | .16 |
| MRSA-152 | .26 | .26 | .11 |
| CaffeNet iter 1 | .32 | .29 | .39 |
| CaffeNet iter 50K | .15 | .19 | .16 |
| CaffeNet iter 310K | .16 | .12 | .18 |
| Gabor Decomposition | .26 | .27 | **.48** |
| Steerable Pyramid | .24 | .32 | .25 |

Table 6: Background context for Shape. Shown is the Spearmann correlation coefficient (SROCC) of perceptual data vs. model-based MI prediction across shapes (i.e. considering all shapes rather than only Easy vs. Hard; note that the original robust finding the superiority of the Easy shape). Perceptual data from Weisstein & Harris (1974), where "Day 1" and "Days 2-4" (averaged) are for the reduced-masking condition depicted in their Figure 3.)

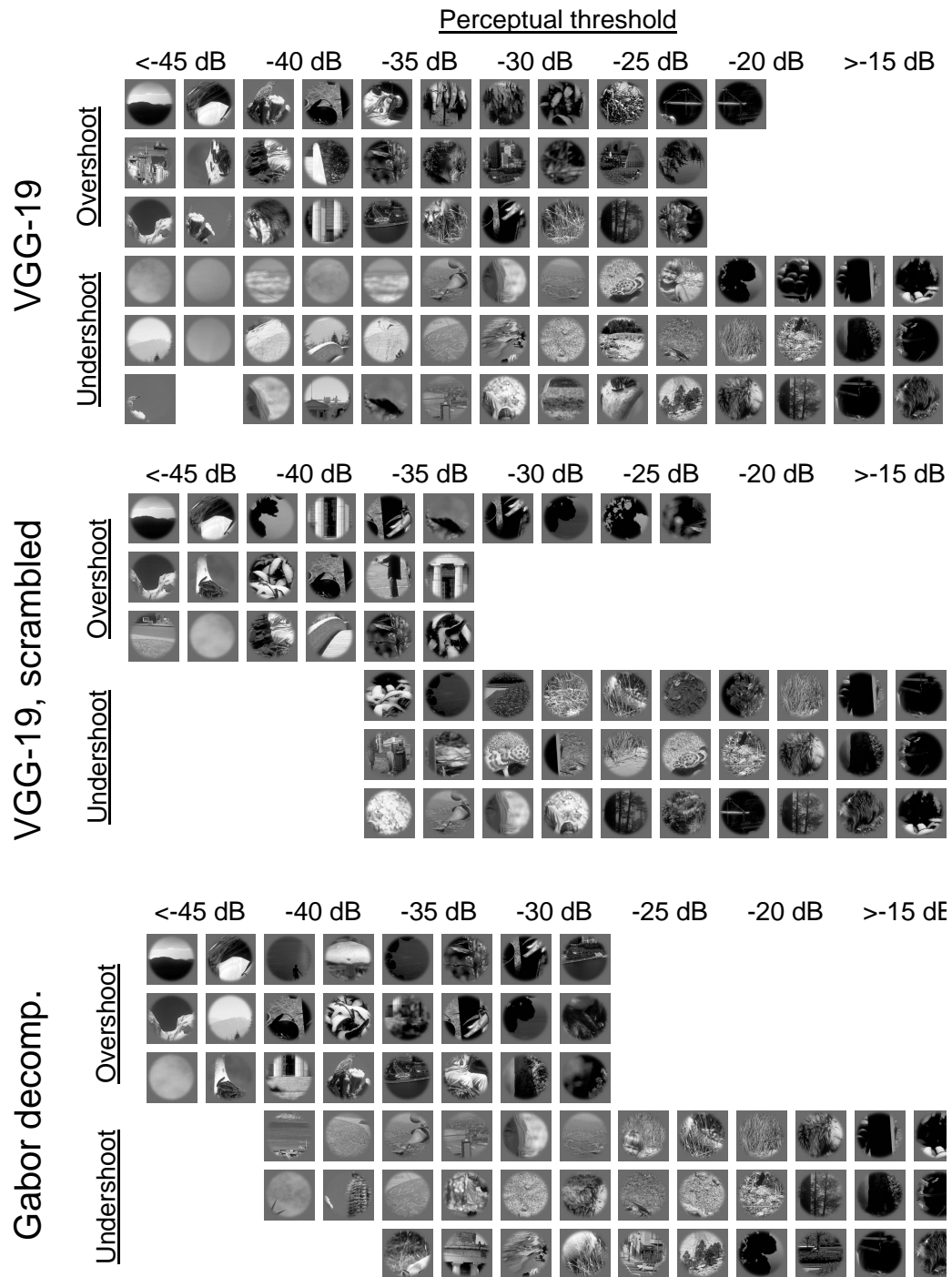

Figure 6: Images where predicted threshold is too high ("Overshoot", where perturbation saliency is better than predicted) or too low ("Undershoot"), considered from several perceptual threshold ranges (±2 dB of shown number). Some images are reproduced from Figure 1.

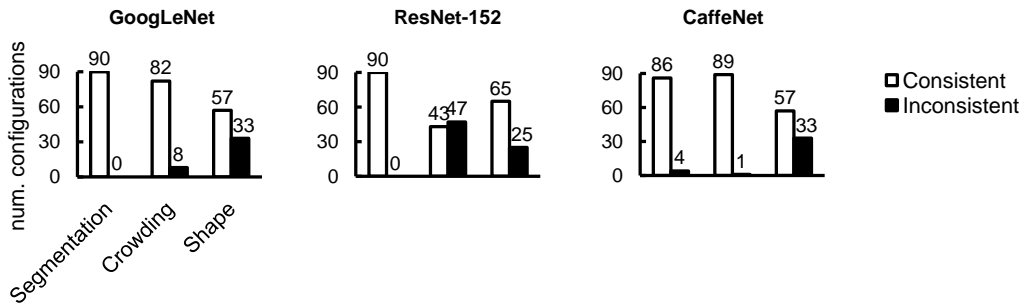

Figure 7: Background context for different DNN models (following figure 2).

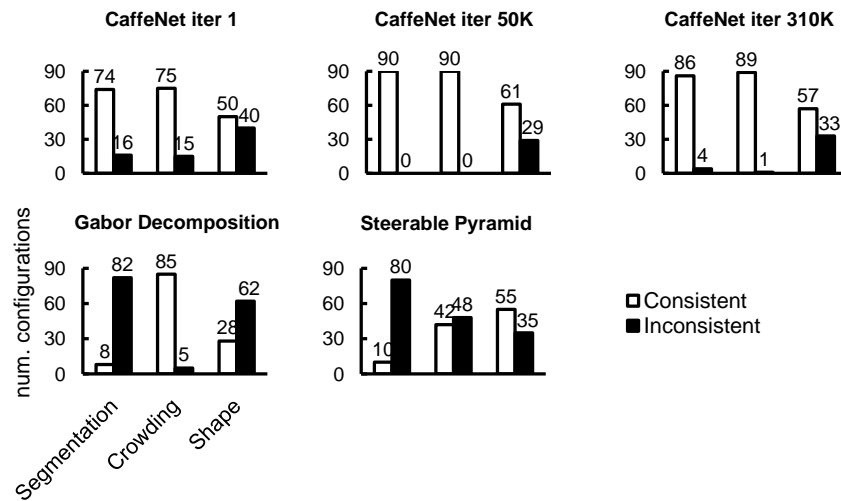

Figure 8: Background context for baseline DNN models (following figure 2). "CaffeNet iter 310K" is reproduced from Figure 7.

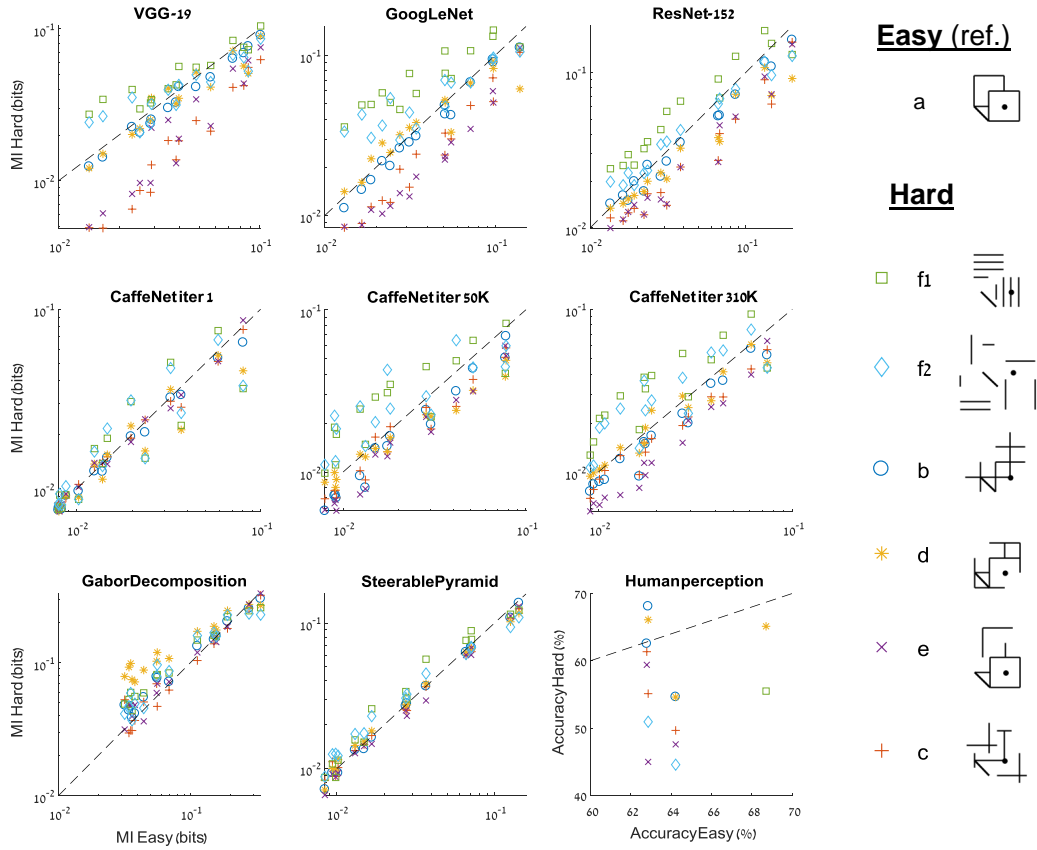

Figure 9: Background context for Shape. Shown for each model is the measured MI for the six "Hard" shapes as a function of the MI for the "Easy" shape. The last panel shows an analagous comparison measured in human subjects by Weisstein & Harris (1974). A data point which lies below the dashed diagonal indicates a configuration for which discriminating line location is easier for the Easy shape compared with the relevant Hard shape.

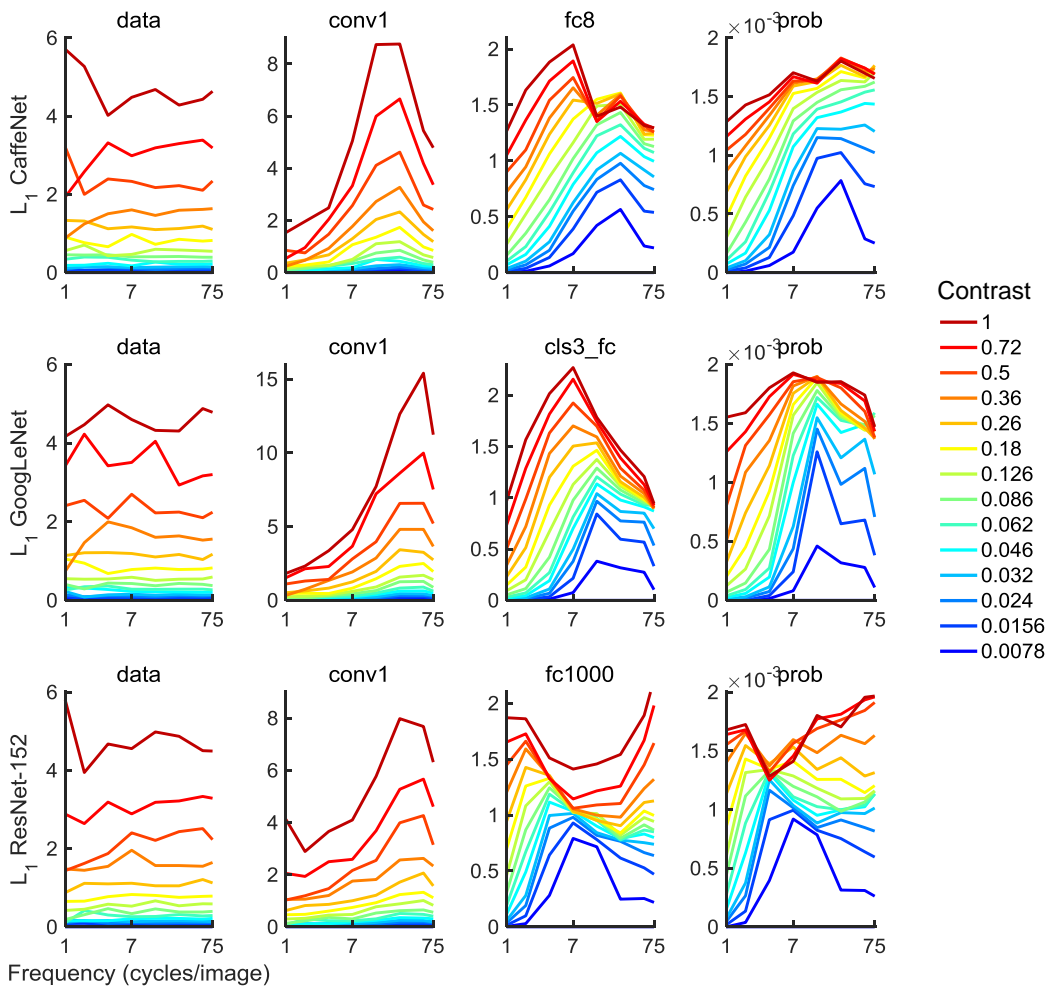

Figure 10: Contrast sensitivity (following Figure 3) for DNN architectures CaffeNet, GoogLeNet, and ResNet-152.

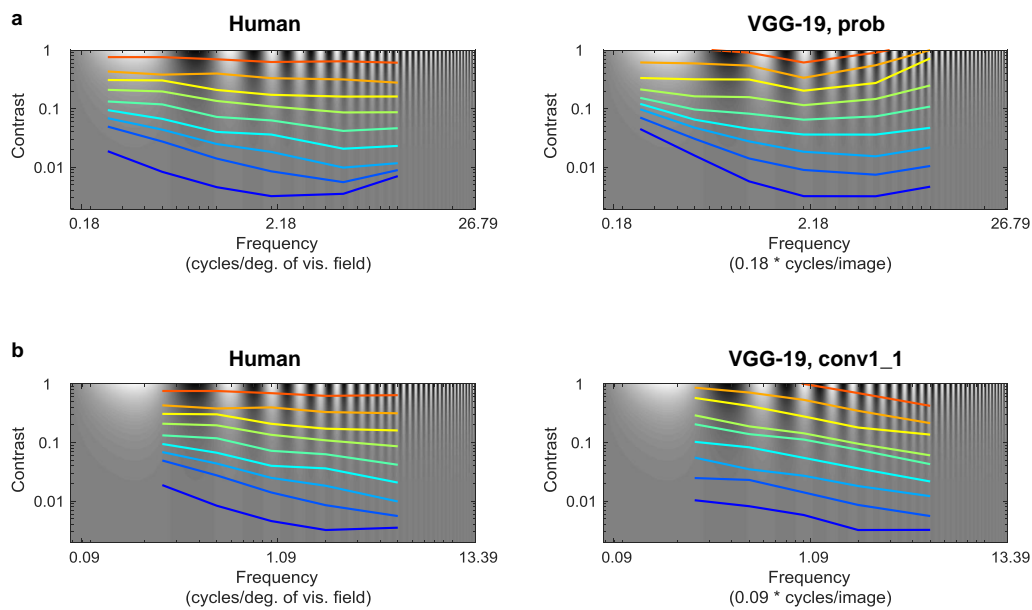

Figure 11: Comparison of contrast sensitivity. Shown are iso-output curves, for which perceived contrast is the same (Human), or for which the $L_1$ change relative to a gray image is the same (DNN model VGG-19). To obtain a correspondence between human frequency values (given in cycles per degree of visual field) to DNN frequency values (given in cycles per image), a scaling was chosen such that the minima of the blue curve is given at the same frequency value. Human data is for subject M.A.G. as measured by Georgeson & Sullivan (1975).

## 8 APPENDIX: EXPERIMENTAL SETUP

### 8.1 DNN MODELS

To collect DNN computation snapshots, we used MATLAB with MatConvNet version 1.0-beta20 (Vedaldi & Lenc, 2015). All MATLAB code will be made available upon acceptance of this manuscript. The pre-trained DNN models we have used are: CaffeNet (which is a variant of AlexNet provided in Caffe, Jia et al., 2014), GoogLeNet (Szegedy et al., 2014), VGG-19 (Simonyan & Zisserman, 2014), and ResNet-152 (He et al., 2015). The models were trained on the same ImageNet LSVRC. The CaffeNet model was trained using Caffe with the default ImageNet training parameters (stopping at iteration $310,000$) and imported into MatConvNet. For the GoogLeNet model, we used the imported pre-trained reference-Caffe implementation. For VGG-19 and ResNet-152, we used the imported pre-trained original versions. In all experiments input image size was $224 \times 224$ or $227 \times 227$.

### 8.2 BASELINE MODELS

As baselines to compare with pre-trained DNN models, we consider: (a) a multiscale linear filter bank of Gabor functions, (b) a steerable-pyramid linear filter bank (Simoncelli & Freeman, 1995), (c) the VGG-19 model for which the learned parameters (weights) were randomly scrambled within layer, and (d) the CaffeNet model at multiple time points during training. For the Gabor decomposition, the following Gabor filters were used: all compositions of $\sigma = \{1, 2, 4, 8, 16, 32, 64\}$px, $\lambda = \{1, 2\} \cdot \sigma$, orientation= $\{0, \pi/3, 2\pi/3, \pi, 4\pi/3, 5\pi/3\}$, and phase= $\{0, \pi/2\}$.

### 8.3 IMAGE PERTURBATION EXPERIMENT

The noiseless images were obtained from Alam et al. (2014). In main text, "image scale" refers to percent coverage of DNN input. Since size of original images ($149 \times 149$) is smaller than DNN input of ($224 \times 224$) or ($227 \times 227$), the images were resized by a factor of $1.5$ so that 100% image scale covers approximately the entire DNN input area.

Human psychophysics and DNN experiments were done for nearly identical images. A slight discrepancy relates to how the image is blended with the background in the special case where the region where noise is added has no image surround at one or two side. In these sides (which depend on the technical procedure with which images were obtained, see Alam et al., 2014), the surround blending here was hard, while the original was smooth.

### 8.4 BACKGROUND CONTEXT EXPERIMENT

#### 8.4.1 SEGMENTATION

The images used are based on the Texture Discrimination Task (Karni & Sagi, 1991). In the variant considered here (Pinchuk-Yacobi et al., 2015), subjects were presented with a grid of lines, all of which were horizontal, except two or three that were diagonal. Subjects discriminated whether the arrangement of diagonal lines is horizontal or vertical, and this discrimination was found to be more difficult when the central line is horizontal rather than diagonal ("Hard" vs. "Easy" in Figure 2a). To limit human performance in this task, two manipulations were applied: (a) the location of each line in the pattern was jittered, and (b) a noise mask was presented briefly after the pattern. Here we only retained (a).

A total of 90 configurations were tested, obtained by combinations of the following alternatives:

- Three scales: line length of 9, 12.3, or 19.4 px (number of lines co-varied with line length, see Figure 12).
- Three levels of location jittering, defined as a multiple of line length: $\{1, 2, 3\} \cdot 0.0625 \cdot l$ px, where $l$ is the length of a line in the pattern. Jittering was applied separately to each line in the pattern.
- Ten locations of diagonal lines: center, random, four locations of half-distance from center to corners, four locations of half-distance from center to image borders.

For each configuration, the discriminated arrangement of diagonal lines was either horizontal or vertical, and the central line was either horizontal or diagonal (i.e. hard or easy).

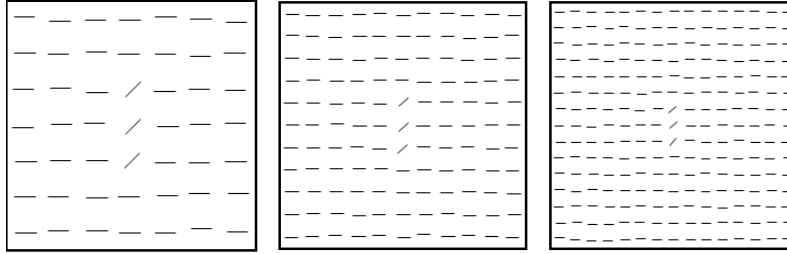

Figure 12: Pattern scales used in the different configurations of the Segmentation condition. Actual images used were white-on-black rather than black-on-white.

### 8.4.2   CROWDING

The images used are motivated by the crowding effect (Livne & Sagi, 2007; Pelli et al., 2004).

A total of 90 configurations were tested, obtained by combinations of the following alternatives:

- Three scales: font size of 15.1, 20.6, or 32.4 px (see Figure 13).
- Three levels of discriminated-letter location jittering, defined as a multiple of font size: $\{1, 2, 3\} \cdot 0.0625 \cdot l$ px, where $l$ is font size. The jitter of surround letters (M, N, S, and T) was fixed (i.e. the background was static).
- Ten locations: center, random, four locations of half-distance from center to corners, four locations of half-distance from center to image borders.

For each configuration, the discriminated letter was either A, B, C, D, E, or F, and the background was either blank (easy) or composed of the letters M, N, S, and T (hard).

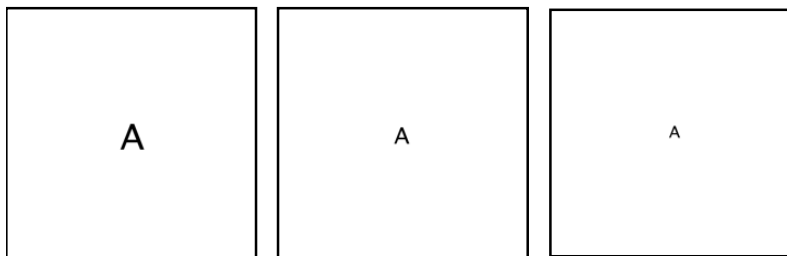

Figure 13: Pattern scales used in the different configurations of the Crowding condition. Actual images used were white-on-black rather than black-on-white.

### 8.4.3   SHAPE

The images used are based on the object superiority effect by Weisstein & Harris (1974), where discriminating a line location is easier when combined with surrounding lines a shape is formed.

A total of 90 configurations were tested, obtained by combinations of the following alternatives:

- Three scales: discriminated-line length of 9, 15.1, or 22.7 px (see Figure 14).
- Five levels of whole-pattern location jittering, defined as a multiple of discriminated-line length: $\{1, 2, 5, 10, 15\} \cdot 0.0625 \cdot l$ px, where $l$ is the length of the discriminated line.

- Six "hard" background line layouts (patterns *b-f* of their Figure 2 and the additional pattern *f* of their Figure 3 in Weisstein & Harris, 1974). The "easy" layout was always the same (pattern *a*).

For each configuration, the line whose location is discriminated had four possible locations (two locations are shown in Figure 2c), and the surrounding background line layout could compose a shape (easy) or not (hard).

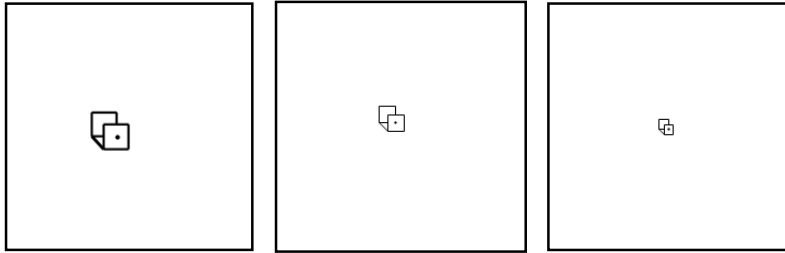

Figure 14: Pattern scales used in the different configurations of the Shape condition. Actual images used were white-on-black rather than black-on-white.

## 8.5 CONTRAST SENSITIVITY EXPERIMENT

Used images depicted sine gratings at different contrast, spatial frequency, sine phase, and sine orientation combinations.

