# Peer review of "Human perception in computer vision"

_ICLR 2017 — rejected_

[Official Review · AnonReviewer3 · rating 7 · confidence 3 · 16 Dec 2016 (modified: 18 Jan 2017)]
**Updated Review**

The paper reports several connections between the image representations in state-of-the are object recognition networks and findings from human visual psychophysics:
1) It shows that the mean L1 distance in the feature space of certain CNN layers is predictive of human noise-detection thresholds in natural images.
2) It reports that for 3 different 2-AFC tasks for which there exists a condition that is hard and one that is easy for humans, the mutual information between decision label and quantised CNN activations is usually higher in the condition that is easier for humans.
3) It reproduces the general bandpass nature of contrast/frequency detection sensitivity in humans. 

While these findings appear interesting, they are also rather anecdotal and some of them seem to be rather trivial (e.g. findings in 2). To make a convincing statement it would be important to explore what aspects of the CNN lead to the reported findings. One possible way of doing that could be to include good baseline models to compare against. As I mentioned before, one such baseline should be reasonable low-level vision model. Another interesting direction would be to compare the results for the same network at different training stages.

In that way one might be able to find out which parts of the reported results can be reproduced by simple low-level image processing systems,  which parts are due to the general deep network’s architecture and which parts arise from the powerful computational properties (object recognition performance) of the CNNs.

In conclusion, I believe that establishing correspondences between state-of-the art CNNs and human vision is a potentially fruitful approach. However to make a convincing point that found correspondences are non-trivial, it is crucial to show that non-trivial aspects of the CNN lead to the reported findings, which was not done. Therefore, the contribution of the paper is limited since I cannot judge whether the findings really tell me something about a unique relation between high-performing CNNs and the human visual system.

UPDATE:

Thank you very much for your extensive revision and inclusion of several of the suggested baselines. 
The results of the baseline models often raise more questions and make the interpretation of the results more complex, but I feel that this reflects the complexity of the topic and makes the work rather more worthwhile. 

One further suggestion: As the experiments with the snapshots of the CaffeNet shows, the direct relationship between CNN performance and prediction accuracy of biological vision known from Yamins et al. 2014 and Cadieu et al. 2014 does not necessarily hold in your experiments. I think this should be discussed somewhere in the paper.

All in all, I think that the paper now constitutes a decent contribution relating state-of-the art CNNs to human psychophysics and I would be happy for this work to be accepted.

I raise the my rating for this paper to 7.

[Official Review · AnonReviewer1 · rating 6 · confidence 3 · 18 Dec 2016]
**Review of "HUMAN PERCEPTION IN COMPUTER VISION"**

This paper compares the performance, in terms of sensitivity to perturbations, of multilayer neural networks to human vision.  In many of the tasks tested, multilayer neural networks exhibit similar sensitivities as human vision.  

From the tasks used in this paper one may conclude that multilayer neural networks capture many properties of the human visual system.  But of course there are well known adversarial examples in which small, perceptually invisible perturbations cause catastrophic errors in categorization, so against that backdrop it is difficult to know what to make of these results.  That the two systems exhibit similar phenomenologies in some cases could mean any number of things, and so it would have been nice to see a more in depth analysis of why this is happening in some cases and not others.  For example, for the noise perturbations described in the the first section, one sees already that conv2 is correlated with human sensitivity.  So why not examine how the first layer filters are being combined to produce this contextual effect?  From that we might actually learn something about neural mechanisms.

Although I like and am sympathetic to the direction the author is taking here, I feel it just scratches the surface in terms of analyzing perceptual correlates in multilayer neural nets.

[Official Review · AnonReviewer2 · rating 6 · confidence 4 · 20 Dec 2016]
**Review of "Human Perception in Computer Vision"**

The author works to compare DNNs to human visual perception, both quantitatively and qualitatively. 

Their first result involves performing a psychophysical experiment both on humans and on a model and then comparing the results (actually I think the psychophysical data was collected in a different work, and is just used here).   The specific psychophysical experiment determined, separately for each of a set of approx. 1110 images, what the noise level of additive noise would have to be to make a just-noticeable-difference for humans in discriminating the noiseless image from the noisy one.   The authors then define a metric on neural networks that allows them to measure what they posit might be a similar property for the networks.  They then correlate the pattern of noise levels between neural networks that the humans.    Deep neural networks end up being much better predictors of the human pattern of noise levels than simpler measure of image perturbation (e.g. RMS contrast).  

A second result involves comparing DNNs to humans in terms of their pattern errors in a series of highly controlled experiments using stimuli that illustrate classic properties of human visual processing -- including segmentation, crowding and shape understanding.  They then used an information-theoretic single-neuron metric of discriminability to assess similar patterns of errors for the DNNs.   Again, top layers of DNNs were able to reproduce the human patterns of difficulty across stimuli, at least to some extent. 

A third result involves comparing DNNs to humans in terms of their pattern of contrast sensitivity across a series of sine-grating images at different frequencies.  (There is a classic result from vision research as to what this pattern should be, so it makes a natural target for comparison to models.)   The authors define a DNN correlate for the propertie in terms of the cross-neuron average of the L1-distance between responses to a blank image and responses to a sinuisoid of each contrast and frequency.   They then qualitatively compare the results of this metric for DNNs models to known results from the literature on humans, finding that, like humans, there is an apparent bandpass response for low-contrast gratings and a mostly constant response at high contrast.  

Pros:
    * The general concept of comparing deep nets to psychophysical results in a detailed, quantitative way, is really nice.   

    * They nicely defined a set of "linking functions", e.g. metrics that express how a specific behavioral result is to be generated from the neural network.  (Ie. the L1 metrics in results 1 and 3 and the information-theoretic measure in result 2.)   The framework for setting up such linking functions seems like a great direction to me. 

    * The actual psychophysical data seems to have been handled in a very careful and thoughtful way.   These folks clearly know what they're doing on the psychophysical end.  


Cons:
    * To my mind, the biggest problem wit this paper is that that it doesn't say something that we didn't really know already.   Existing results have shown that DNNs are pretty good models of the human visual system in a whole bunch of ways, and this paper adds some more ways.    What would have been great would be: 
         (a) showing that they metric of comparison to humans that was sufficiently sensitive that it could pull apart various DNN models, making one clearly better than the others. 
         (b) identifying a wide gap between the DNNs and the humans that is still unfilled.   They sort of do this, since while the DNNs are good at reproducing the human judgements in Result 1, they are not perfect -- gap is between 60% explained variance and 84% inter-human consistency.    This 24% gap is potentially important, so I'd really like to see them have explored that gap more -- e.g. (i) widening the gap by identifying which images caused the gap most and focusing a test on those, or (ii) closing the gap by training a neural network to get the pattern 100% correct and seeing if that made better CNNs as measured on other metrics/tasks. 

In other words, I would definitely have traded off not having results 2 and 3 for a deeper exploration of result 1.    I think their overall approach could be very fruitful, but it hasn't really been carried far enough here. 

   * I found a few things confusing about the layout of the paper.  I especially found that the quantitative results for results 2 and 3 were not clearly displayed.   Why was figure 8 relegated to the appendix?  Where are the quantifications of model-human similarities for the data shown in Figure 8?  Isn't this the whole meat of their second result?   This should really be presented in a more clear way.    

    * Where is the quantification of model-human similarity for the data show in Figure 3?  Isn't there a way to get the human contrast-sensitivity curve and then compare it to that of models in a more quantitively precise way, rather than just note a qualitative agreement?   It seems odd to me that this wasn't done.

[Author Response · Ron Dekel · 15 Jan 2017 (modified: 17 Jan 2017)]
**Edits**

Jan 16-17, 2017:
- Edited the hypothesis about "Overshoot" and "Undershoot" inconsistency with perception (result 1).
- Added the prediction quality of perceptual threshold as a function of layer for model ResNet-152 .

Jan 15, 2017:
-	Added results for three baseline models: two linear filter banks (Gabor decomposition and steerable pyramid) and VGG-19 with scrambled weights.
-	Added results for CaffeNet model at several snapshots during training.
-	Added human data for contrast sensitivity (figure 3 results).
-	Added configurations for context experiments (figure 2 results). Now there are 90 configurations per CNN architecture for Segmentation, Crowding, and Shape.
-	Cosmetics and minor corrections.

[Final Decision · Program Chairs · 06 Feb 2017]
**ICLR committee final decision**

I think the reviewers evaluated this paper very carefully and were well balanced. The reviewers all agree that the presented comparison between human vision and DNNs is interesting. At the same time, none of the reviewers would strongly defend the paper. As it stands, this work seems a little too premature for publication as the analysis does not go too much beyond what we already know. We encourage the authors to deepen the investigation and resubmit.